# Modeling Root Exudate Accumulation Gradients to Estimate Net Exudation Rates by Peatland Soil Depth

**DOI:** 10.3390/plants10010106

**Published:** 2021-01-06

**Authors:** Cameron Proctor, Yuhong He

**Affiliations:** 1School of the Environment, University of Windsor, 401 Sunset Ave, Windsor, ON N9B 3P4, Canada; 2Department of Geography, Geomatics and Environment, University of Toronto Mississauga, 3359 Mississauga Road, William G. Davis Bldg, Mississauga, ON L5L 1C6, Canada; yuhong.he@utoronto.ca

**Keywords:** root exudation, rhizosphere, simulation model, peatlands

## Abstract

Root exudates accumulate as a radial gradient around the root, yet little is known about variability at the individual root level. Vertical gradients in soil properties are hypothesized to cause greater accumulation of exudates in deeper soil through hindering diffusion, increasing sorption, and decreasing mineralization. To this end, a single root exudation model coupling concentration specific exudation and depth dependent soil properties was developed. The model was parameterized for a peatland ecosystem to explore deposition to the methanogen community. Numerical experiments indicate that exudates accumulated to a greater extent in deeper soil, albeit the effect was solute specific. Rhizosphere size for glucose doubled between the 10 and 80 cm depths, while the rhizoplane concentration was 1.23 times higher. Root influx of glucose increased from 1.431 to 1.758 nmol cm^−1^ hr^−1^, representing a recapture efficiency gain of 15.74% (i.e., 69.06% versus 84.8%). Driven by increased root influx, overall net exudation rates of select sugars and amino acids varied by a factor two. Model sensitivity analysis revealed that soil depth and root influx capability are key determinants of the rhizoplane concentration and subsequently net exudation, which determines whether effluxed compounds escape the root oxic shell and are available to the methanogen community.

## 1. Introduction

The loss of carbon (C) from roots to soil is an important, but poorly understood process relative to other pathways in the terrestrial C cycle [1,2]. At the rhizoplane, C loss via exudation ranges from 5–10% of net fixed C [3], yet considerable variation in exudate magnitude and composition occurs between species [4] and in relation to soil conditions (e.g., nutrient availability [5], temperature [6], and oxygen concentration [7]). Since root-borne C strongly interacts with the soil microbial community through symbiotic to pathogenic interactions, roots have developed some ability to control their efflux which consists of primary metabolites (i.e., organic acids, amino acids, sugars), secondary metabolites (plant specialized metabolites), and proteins. Although exudation of C is partially unintentional (passive efflux), efflux of secondary metabolites has been linked to microbial chemo-attraction, biological communication, and shaping the soil microbial community [8] and is therefore believed to be plant regulated (active efflux). Albeit the majority of exuded C are primary metabolites [9], concordance between internal root composition and rhizosphere composition is poor [10], suggesting the mechanisms involved in exudation are highly selective.

The assumption that higher internal root concentrations leads to higher exudation rates neglects that exudation is the net release of C compounds with efflux and influx components [11]. Asymmetries between efflux and influx may explain why select C compounds occur in high proportion in root extracts, yet are a minor constituent of exudates [12]. Studies of ryegrass, rape, clover, and alfalfa document that exudation rates and the soil concentration of the amino acids glycine and serine are higher than their internal root concentration suggests [12,13,14]. Wheat roots were highly proficient at recapturing nitrogen (N) compounds effluxed from roots, with influx rates exceeding efflux rates for 21 out of 45 15N-labeled compounds [15]. Net influx may thereby explain the radial depletion zones of amino acids that have been documented in the soil solution of *Zea mays* grown in rhizoboxes fitted with micro-suction-cups [16].

Root efflux of primary metabolites is regarded as passive leakage to which the plant has little control over. Root permeability that permits uptake subsequently permits diffusion-based efflux [17] based on the concentration gradient between the root cell cytoplasmic concentration (mM concentration) and the rhizosphere (µmol concentration), where low soil concentration is continually maintained via diffusion away from the rhizoplane and microbial mineralization [18]. While roots display little ability to down-regulate their passive C efflux, active exudation involving up-regulating existing diffusion mechanisms [2] can be triggered under non-optimal conditions such as during exposure to metals and strongly charged substances [19], or due to low nutrient availability [20]. Evidence for the active release of organic acid exudates in response to soil conditions has been long documented [21]. Recapturing the organic acids would be energetically unfavorable due to their charge, hence there is little evidence for organic acid recapture [3]. While influx of negatively charged compounds is hindered, the positive charge on the exterior face of the root draws the compounds out of the cell at higher rates [22]. Root recapture of lost compounds may be energetically or nutritionally favorable despite the energy cost of influx, with several studies noting high influx rates for sugars [3] and amino acids [23]. Hence, the charge of the C compound, coupled with its intrinsic value to plant ecophysiology, dictates whether net exudation is regulated by influx or efflux processes.

Net exudation to the soil results in an accumulation gradient around the root with the highest concentration of exudates at the rhizoplane, decreasing radially until they subside at the boundary of the rhizosphere. Extending the radial distribution thereby spreads the exudate pool over a larger soil volume, reducing the concentration at the rhizoplane where concentration dependent efflux and influx processes occur [2]. Although many species display the capacity to take up amino acids at concentrations as low as 2 µmol, uptake rates are considerable greater at higher concentration, with maximum rates often an order of magnitude greater than typical rhizoplane concentrations [24]. Quantifying the rhizoplane concentration is challenging since it is affected by the net exudation rate, radial diffusion rate, transport with water movement, sorption to the solid phase, and decomposition of exudates by microorganisms, all of which vary with soil vertical gradients in physical, chemical, and biological properties.

Relative to surface soil, deep soil has undergone many additional cycles of microbial processing, stripping it of energy-rich C and enriching the microbial-processed C relative to topsoil. In rice paddies, lowered abundance of plant litter reduced the overall microbial biomass in deeper soil, and further selected for species that specialize in the digestion of soil organic matter [25]. Similarly, microbial biomass of a bare cutover peatland varied from 326 to 281 µg C g dwt^−1^ between the 0–5 cm and 40–45 cm soil depth intervals, while a concordant MicroResp assay documented mineralization rates of isotopically labeled C up to ~2.8 times greater in 0–5 cm than 40–45 cm soil depth intervals [26]. Reduced removal via microbial consumption may enable greater recapture of effluxed compounds, lowering net exudation rates. The effect of topsoil versus subsoil properties on root exudation was investigated using pulse labeled ^14^C tracers of alfalfa in pots filled with either a topsoil (0–30 cm) or subsoil (45–75 cm) [27]. Their research documented root exudation per root mass was lower in subsoil, while the topsoil had a larger accumulation gradient of labeled ^14^C exudates in the rhizosphere and rhizoplane, despite increased microbial utilization. Similarly, a trend of decreasing in-situ root exudation with depth from 33–82 mg C root DW g^−1^ h^−1^ in the topsoil to an average of 5–33 mg C root DW g^−1^ h^−1^ between the 60 and 130 cm depth intervals was noted using a cuvette-based approach [28]. However, empirical observations have many methodological limitations [29].

Replication of soil properties and vertical gradients to empirically investigate the radial distribution of exudates around roots is challenging, necessitating the development of simulation models that differ in approach, complexity, and theoretical underpinnings. Previously, impedance and interfacial resistance terms were used to predict diffusion profiles of non-sorbing and sorbing solutes [30]. Recently, single root exudation models have been developed [31] that include influx-efflux processes [3]. While this model was empirically validated for maize roots grown in solution, the model does not simulate soil property effects on exudation. Conversely, the PARIS-M [32] and Barber–Cushman rhizosphere gradient models [32] focus on soil property influences on accumulation gradients in the rhizosphere with net exudation rate as a constant. Models for the spatiotemporal distribution of secondary metabolites have also been developed, such as the secretion of daidzein from soybean roots [33]. However, there is a paucity of rhizosphere simulation models coupling efflux and influx proportional to the concentration on the rhizoplane with soil property determinants of rhizoplane concentration.

Vertical gradients in the soil properties and their influence on net exudation rates in peatlands is unknown, despite the considerable vertical heterogeneity due to compaction and decomposition. Unlike non-saturated ecosystems, the root systems of peatland plants are partitioned into an oxic and anoxic zone via a fluctuating water table. The ratio of root deposited C mineralized as CO_2_ versus CH_4_ thus depends on the water table depth, the root depth distribution, and whether the soil properties of the inundated root fraction affect net exudation sink strength. Exudation into the anoxic zone in peatlands may relieve methanogen substrate limitation, or accumulation of organic acids may suppress methanogenesis [34], despite acetate typically stimulating CH_4_ production [35]. Methanogenesis operates within a narrow window of environmental and chemical concentration conditions, hence the considerable uncertainty regarding the contribution of root exudates to methane production. Understanding depth variations in net exudate magnitude and the extent of the rhizosphere may offer valuable insights into the development of methanogenesis spatial niches.

The objectives of this paper are to better understand how depth dependent soil properties determine the rhizoplane concentration and influence net exudation of a deep-rooted peatland sedge (*Eriophorum vaginatum)* under saturated conditions. The model focuses on predicting accumulation gradients of effluxed C compounds around an immutable root segment as a function of radial diffusion and removal via sorption to the solid phase and microbial mineralization. This single root approach significantly simplifies the root system and does not account for all factors influencing the rhizosphere accumulation gradients, such as root age, seasonality, and inter-root competition. However, the cylindrical representation matches the root architecture of *E. vaginatum* which is branchless, absent of root hairs, and propagates downwards from the central stem. Internal cell concentrations of the simulated root are kept constant regardless of depth, while soil properties alter in response to soil depth in concordance with observations.

## 2. Results

Simulated soil vertical gradients had considerable variation in bulk density and decreasing porosity and increasing tortuosity with depth (Figure 1), thereby interfering with radial diffusion of exudates. Increased sorption to the denser solid phase further reduced radial diffusion, albeit the sorption retardation factor depended on the type of solute and its concentration. In high binding compounds such as oxalate, the sorption retardation factors influence on diffusion was larger than porosity. In terms of mineralization rates, soils became cooler and featured a smaller microbial biomass with depth, reducing mineralization.

### 2.1. Rhizosphere Accumulation Gradients by Depth

For all sugars and amino acids, in deeper soil, a greater amount of exudates were retained in the rhizosphere, generally increasing the concentration present, including the rhizoplane (Figure 2). For organic acids, the effect was less pronounced. In deeper soil, the heightened accumulated exudates were radially dispersed amongst a larger soil volume, increasing the rhizosphere size. Defining the rhizosphere at an arbitrary threshold of 0.01 nmol cm^−3^, the extent of the rhizosphere for glucose increased from 7.16 mm at 10 cm to 14.72 mm at 80 cm depths. Furthermore, concentrations at 10 mm were over 70 times greater at 80 cm depth, albeit the difference in concentration shrank closer to the rhizoplane, which was only ~1.23 times greater.

The higher concentration of exudates in the rhizosphere resulted in increased sorption. Total sorbed pool size for all cylinder increased by an order of magnitude from 0.508 to 3.874 nmol cm^−3^, or 2.00% and 6.90% of the solute pool size. At the rhizoplane, the sorption pool size increased by a factor of 4.24 times between depths of 10 and 80 cm, and the amount of solute removed via sorption per time step increased considerably on account of the larger solid phase. However, removal by sorption was concentration dependent. At equilibrium, the solute concentration in the cylinders remains constant; hence, removal via sorption was negligible in spite the higher capacity for solute binding in deep soil.

While sorption increased in deeper soil, mineralization for all soil cylinders decreased up to 6 times between 10 and 80 cm depths. Slower mineralization rates permitted a larger accumulation of exudates in the rhizoplane, which were subsequently dispersed and retained for a longer period, producing an egalitarian distribution of exudates, that tampered diffusion between adjacent soil cylinders. At 80 cm depth, the effective diffusion rate was 63.16% that of 10 cm and the amount of solute diffused from the rhizoplane to the adjacent soil cylinder decreased from 0.0203 to 0.0105 nmol s^−1^ due to the lower diffusion rate in conjunction with the lower solute concentrating difference in the outwards soil cylinder.

Rhizoplane concentrations were higher in deeper soil (e.g., rhizoplane concentration for glucose 1.23 times higher at 80 cm than 10 cm) (Figure 3a). Higher solute concentration on the rhizoplane had downstream implication for exudation efflux and influx. The gradient between the cell cytoplasm (µmol cm^−3^) and the soil solution (nmol cm^−3^) was so stark that minor increases in the soil had a negligible impact of efflux. Root exudate loss was 2.07320 versus 2.07314 nmol cm^−1^ hr^−1^ between 10 and 80 cm depths. However, roots uptake ability has been estimated as high as ~80% for glucose and ~50% for malate in Maize roots [3]. Influx increased with depth, from 1.431 to 1.758 nmol cm^−1^ hr^−1^, which represented a recapture efficiency gain of 15.74% (i.e., 69.06% versus 84.8%). Because the recapture efficiency of exuded labile C was high, net exudation was drastically reduced to less than 50% its rate at 10 cm (Figure 3b). Driven by increased uptake, net exudation was reduced from 0.641 to 0.315 nmol cm^−1^ hr^−1^ between the 10 and 80 cm depths.

### 2.2. Compound Specific Exudation

Root exudates are a cocktail of low molecular weight (LMW) C compounds with considerable heterogeneity in compound properties. Simulation results show that accumulation gradients vary considerably between C compounds (Figure 4). Consequently, solute properties exerted considerable influence on the final rhizosphere concentration with downstream influences on exudation efflux and influx. Overall, net exudation rates were up to two orders of magnitude different between C compounds. Net exudation of glycine at 40 cm depth was 0.31 nmol cm^−1^ hr^−1^ in comparison to 7.98 nmol cm^−1^ hr^−1^ for tartarate. The lower net exudation of amino acids was mainly due to their low root cytoplasm concentration and high recapture efficiency. The higher rhizoplane concentration in deeper soil resulted in greater recapture of sugars and amino acids (Figure 5a,b).

Conversely, negatively charged solutes such as tartarate are drawn out of the root at greater rates than non-charged solutes [3]. Exuded negatively charged compounds thereby have little influx ability given that the root interior maintains the same negative charge. As tartarate has a net negative charge of −2, its efflux rate is greater than solutes with comparable cell cytoplasm concentration (e.g., glucose: −1.48 nmol cm^−1^ hr^−1^ (37.4 µmol cm^−3^), tartarate: 7.98 nmol cm^−1^ hr^−1^ (59.5 µmol cm^−3^)). More importantly, whereas non-charged C compounds can be recaptured, and are thus regulated by influx mechanisms, organic acids such as tartarate are only regulated by efflux processes. Differences in net exudation between organic acids are the result of observed differences in cell cytoplasm concentration, and rates are generally invariable with depth (Figure 5c).

In addition to the exudation rate, the size of the rhizosphere has implications for methanogenesis. Larger rhizospheres are more likely to escape oxygenation from the root and reach a threshold concentration that signals microbial movement and proliferation. Unlike net exudation rates, there is a positive relationship between depth and the size of the rhizosphere (Figure 6). Rhizosphere size tended to increase with depth for all C compounds, likely due to lower mineralization rates. Rhizosphere size tended to be <5 mm for amino acids, <15 mm for sugars, and <20 mm for organic acids. Diffusion rate had considerable influence on the size of the rhizosphere, with the rhizosphere size of the slowly diffusing tartarate the lowest of all organic acids in spite of its higher exudation rate. The rhizosphere size was linearly related to depth, albeit the slope of the equation was solute specific. Pooled rhizosphere thickness and depth only had an R^2^ of 0.26. Conversely, linear relationships between these two variables for a single solute were in excess of R^2^ of 0.95.

### 2.3. Sensitivity Analysis

The global sensitivity analysis (*n* = 10,000) revealed the general behavior of the model points towards increasing rhizoplane concentration at lower depths (Figure 7). Linear regression with depth as the independent variable and the rhizoplane concertation as the dependent variable was statistically significant (*p* < 0.01) and had an R^2^ of 0.22. Albeit, there is considerable variability in the simulation results, the general trend towards higher rhizoplane concentration in deeper soil is clear.

The partial rank correlation coefficients for the entire model indicated that soil depth and the root influx capabilities had the strongest influence on the rhizoplane concentration (Figure 8). Neither are unsurprising, as soil depth influences mineralization rates while root exudate influx influences recapture efficiency. Both the maximum influx rate and the Michaelis–Menten coefficient were important in the model, likely as the Michaelis–Menten coefficient is crucial for determining the influx strength at low concentration. Moderately important factors were consistent with the diffusion coefficient and the mineralization rate. These factors are constant with soil depth. Hence, they represent the sensitivity of the model to different solute properties. The Q10 factor and soil sorption capabilities had a minor role in the rhizoplane concentration.

The strongest influential parameters, maximum sorption, maximum uptake rate, and Michaelis–Menten coefficient, remained fairly constant by depth (Figure 9). Interestingly, the mineralization rate and the Q10 soil temperature factor declined in the partial rank correlation coefficient rankings between 10 and 100 cm depths. Their influence likely waned as other factors such as the bulk density and microbial biomass exerted a stronger influence in the deeper soil.

## 3. Discussion

Root exudation regulators have previously been identified, yet their application to estimate exudation rates is rare. Furthermore, few studies have examined the role of vertical gradients of exudation regulators on estimated rates. This simulation model has demonstrated quantitatively the regulation of exudation via soil properties with depth dependence in a peatland environment. Vertical gradients in sugar and amino acid net exudation due to higher rhizoplane accumulation led to higher influx of exudates, suggesting that deeper roots may have lower loss of C. Consequently, the higher accumulation of exudates in the rhizosphere increases the width of the rhizosphere. As the rhizosphere width is independent of the oxic shell, larger rhizosphere widths may enable exudates to escape the oxic shell and become available to the methanogen community. In terms of methanogenesis, the higher accumulation gradient and larger rhizosphere are important consideration for exudates escaping the oxic shell around roots. The results of this model offer insights into root exudation regulation and is possible significance in methanogenesis and emissions at the sediment atmosphere interface.

Loss of labile C compounds via the root exudation pathways have been difficult to quantify empirically given the sizeable methodological limitations [5] and wide range in reported magnitude. The annual flux of C in forests was estimated at 9.4 g C m^−2^ year, or 1.5% of NPP utilizing a simple scaling schema [36]. Meanwhile, upscaled single exudation rates to the ecosystem scale using fine root biomass and the relative dominance of each species was estimated at a flux of 3.1–16.6% of aboveground NPP in tropical rainforests [37]. Determining the validity of these estimates is challenging since C exchanges operate at spatial and temporal scales that are not amenable to investigation and the sampling regime imposes a consistent environment. There are at least 11 variables that influence soil respiration, yet few empirical studies are methodologically capable of simulating all soil properties [38]. The dearth of tools and techniques to observe root exudation without interference is a considerable challenge for estimating the loss of C via the root exudate pathway.

Previous root exudation models range considerably in complexity, focus on single to whole plant simulation, and differ in the breadth of belowground processes incorporated [9]. No single model successfully incorporates all processes at a sophisticated level of complexity. Such parsimony is a reasonable tactic given our current understanding of belowground processes. The model developed in this work was of sufficient complexity to investigate whether all roots have homogenous exudation. The main model limitation is the assumption of static root cytoplasm concentration. Empirical evidence suggests that exudation varies temporally at seasonal timescales [36]. Furthermore, soil water uptake dynamics are crudely represented spatially and temporally in this model. Temporally, root water uptake is not constant as it responds to stomata conduction which is subsequently regulated by a set of climatic variables. Spatially, root water uptake varies with depth with root abundance and the spatial arrangement of roots [39].

Numerical analysis revealed the model was sensitive to solute properties. However, the precise behavior of plant roots and exudates are difficult to predict. For instance, considerable variations in sorption have been noted between different forest soils [40], while mineralization rates are unstable even for the same soil across years [41]. While the preponderance of evidence does suggest the parameter values utilized in this study are correct to within a factor or order of magnitude, the model results are sensitive to their variations. Net exudation rates could vary by a factor of seven for the same solute between the poles of the parameter set in the literature. In particular, exudation rates were sensitive to mineralization rates as this factor is the primary mechanisms for removal of solutes on the rhizoplane. However, regulation of mineralization rates by temperature is well documented [42,43], and long term thermal monitoring data exist to provide a large degree of confidence related to its depth sensitivity. Quantifying the role of microbial biomass is more speculative. Despite studies showing a clear decreasing mineralization rate with depth for structural litter, few studies have examined labile C which may be palatable to a wider microbial community than structurally complex litter [44]. Perhaps the process with the most unknown regarding its depth dependence is the diffusion. Many studies have highlighted the complex nature of the peat solid phase [45]. Converting these measured properties of the solid phase into reductions in the effective diffusion rate is a daunting task. However, the insights gained from this model should provide the next generation of modelers with a deeper understanding of root exudate regulation, specifically the depth dependence on exudation rates which could be incorporated as a simple exponential decay function.

## 4. Materials and Methods

The model is based on a Barber–Cushman approach [46] to simulate accumulation gradients around a cylindrical root ($r_{0}\;$) and rhizosphere soil divided into multiple (n) thin vertical concentric annulus of equal thickness that are considered a homogenous medium with properties reflective of the peat layer depth (Figure 10). Efflux (FE) and influx (FI) between the root and the initial soil annulus (the rhizoplane) are proportional to their concentration gradient, with the root internal concentration held constant for simplicities sake. Net exudation into the rhizoplane occurs when FE>FI. Exudation to the rhizoplane is removed from the soil cylinder pool either by sorption to the soil solid phase (S) or mineralization (M). Remaining exudates exchange with adjacent soil cylinders via diffusion (FD), eventually forming an outward radial accumulation gradient until equilibrium is reached and the rhizosphere (R) has reached its full extent. The exudation rate at equilibrium is considered the depth dependent exudation rate.

### 4.1. Root Exudation Modeling Approach

Net exudation follows the efflux-influx model [3]. Exudation from the root into the initial soil cylinder, $C_{n1},$ is given by:(1)$$\frac{dC_{n1}}{dt}=E_{j}-I_{j},$$
where $E_{j}$ is the efflux (μmol cm^−1^ hr^−1^), and $I_{j}$ is the influx (μmol cm^−1^ hr^−1^) of solute j. Efflux for non-charged compounds is described by the equation:(2)$$E_{j}=AP\left(C_{cyto}-C_{soil}\right),$$
where A is the root area (cm^2^/ cm root length), P is the membrane permeability coefficient (cm hr^−1^), $C_{cyto}$ is the root cell cytoplasm concentration (µmol cm^−3^), and $C_{soil}$ is the soil solution concentration (µmol cm^−3^). Influx of non-charged or weakly charged compounds are concentration dependent and described by Michaelis–Menten kinetics [47] where:(3)$$I_{j}=\frac{I_{max}C_{soil}}{\left(K_{I}+C_{soil}\right)},$$
and $I_{max}$ is the maximum uptake rate (µmol C cm^−1^ root hr^−1^) and $K_{I}$ is the Michaelis–Menten coefficient (µmol cm^−3^).

For ionic compounds, the exterior face of the cell prevents in influx from occurring and draws the negatively charged compounds out of the cell at a greater rate dependent on the compound charge at a cytosol pH of 7~7.2. The net flux density equation [48] is used to describe the flux from the roots to the soil as:(4)$$\frac{dC_{n1}}{dt}=A\left(\frac{PZE_{m}F}{RT}\right)\left(\frac{1}{e^{ZE_{m}F/RT}-1}\right)\left(C_{soil\left(j\right)}-C_{cyto\left(j\right)}e^{ZE_{m}F/RT}\right),$$
and where $E_{m}$ is the membrane potential (−120 mV), F is the faraday constant (9.649 × 10^4^ J mol^−1^ V^−1^), R as the gas constant 8.3143 (J mol^−1^ K^−1^), T as the temperature in Kelvin (K), and Z is the charge of the solute is solution.

Flux of charged and non-charged exudates depend on the permeability of the root, which is solute specific due to variations in molecular size, and polarity [49]. However, the interpretation of permeability coefficients from existing experimental data has proven difficult, inhibiting asserting solute specific coefficients with confidence. Therefore, membrane permeability will be assumed invariable within the sugar, organic acid, and amino acid exudate classes [50], with the sugars coefficient of 1.15 × 10^−4^ cm hr^−1^, a higher coefficient organic acids of 4.32 × 10^−4^ cm hr^−1^ based on malate exudation from wheat root tips [3,50,51]. Permeability values for amino acids were not available, hence permeability was assumed the same as sugars as amino acids as they are mainly non-charged.

Model parameterization is based on the soil properties of Mer Bleue, Canada, a raised ombrotrophic bog 15 km east of Ottawa, Ontario (45.41° N, 75.52° W). Root parameters such as root area and internal root concentrations were based on [34] adjusted for units of surface area. *E. vaginatum* root cellular concentrations of glucose and malate are 44.7 and 27.4 µmol cm^−3^, respectively, which are comparable to 40 and 0.5 µmol cm^−3^ for maize roots [3].

### 4.2. Diffusion Modeling Approach ($F_{D}$)

Solute movement in the soil is based on Fick’s law of diffusion based on the difference in solute concentration between adjacent soil cylinders. Aside from the concentration difference, the rate of flux depends upon the solute specific diffusion coefficient, whose values are available in the literature based on empirically measured diffusion in pure water. Solute transport along the inner boundary between two soil cylinders is given by Fick’s Law [52] as:(5)$$F_{D}\left(i,j\right)=-D_{e}\left(i,j\right)\frac{\Delta C_{i,j}}{dr},$$
where $\Delta C_{i,j}$ is the concentration gradient for solute j between soil cylinders i and the solute concentration at radius r in the adjacent soil cylinder, and $D_{e}\left(i,j\right)$ is the effective diffusion coefficient for solute j in soil layer i. Diffusion coefficients for the movement of individual solutes in pure water are modified due to the introduction of a solid phase which reduces the liquid phase space to conducts water [53] and slows the rate of diffusion since the diffusion pathway elongates from a straight line based on the morphology of the pore space in three dimensions [54]. Diffusion rates are further attenuated by the capacity of the solid phase to physio-chemically bind C compounds. Thus, the effective diffusion of a sorbing solute in saturated media is represented as the fraction of the volume amenable to solute flow, (ϵ(i) <1), tortuosity of the diffusion pathway, (τ(i)>1), and a sorption retardation factor, (R(i,j)>1) as:(6)$$D_{e}\left(i,j\right)=\frac{D_{\infty \left(j\right)}\epsilon \left(i\right)}{\tau {\left(i\right)}^{2}R\left(i,j\right)},$$
where $D_{\infty \left(j\right)}$ (cm^2^ s^−1^) is the diffusion coefficient of solute j in pure liquid water.

Compression and decomposition increase the mass of the solid phase per unit volume in deeper peat [55]. The faction amenable to solute flow is based on the porosity (∅) of soil layer i, where porosity is derived from bulk density [56]. Porosity [57] is the difference between a baseline specific gravity of peat $g_{s}$ (g cm^−3^) minus the bulk density as:(7)$$\epsilon \left(i\right)=\emptyset \left(i\right)=\frac{\left(g_{s}-\rho \left(i\right)\right)}{g_{s}},$$
where the baseline specific gravity of peat is 1.5 g cm^−3^, as has been utilized by [57] and [58] for hydrological modeling at Mer Bleue. Bulk density is modeled as a power function [58] which is a power function for measured bulk density, ρ (g cm^−3^), by depth, d (cm) as:(8)$$\rho \left(d\right)=a_{\rho }{\left(d\right)}^{b_{\rho }},$$
where at Mer Bleue, $a_{\rho }=0.0107$, $b_{\rho }=0.567$, equaling a bulk density increase from 0.0107 g cm^−3^ at the surface to an order of magnitude larger at 0.109 g cm^−3^ at 60 cm depth.

Depth further influences the pore geometry, shifting to fewer macropores, decreased average pore size, and the number of dead end flow paths increase [59,60]. As pore spaces close, the active porosity can be considerably less than the interparticle pore spaces actively transmitting solutes [61]. Tortuosity of the peat with depth is estimated from the porosity based on the Archie’s power law [62]:(9)$$\tau \left(d\right)=\;\emptyset {\left(d\right)}^{-m},$$
where a value of m=2.3 provided the best fit for measured values at three northern Canadian peatland sites [61]. The retardation factor accounts for the interactions of the solute with the sorbing solid phase. As sorption is concentration dependent, the retardation factor increases until saturation is reached. This approach assumes sorption operates many times quicker than diffusion. The retardation factor [63] is:(10)$$R\left(i,j\right)=1+\frac{Csoil\left(i,j\right)+S_{j}\left(Csoil\left(i,j\right)\right)\rho_{i}}{Csol\left(i,j\right)},$$
where $\;S_{j}$ is the slope of the absorption isotherm for solute j at the concentration in the diffusing soil cylinder  Csol(i,j), and $\rho_{i}$ is the bulk density of layer i.

### 4.3. Sorption Modeling Approach (S)

Sorption is often related to soil mineral and metal content, yet organic acid sorption in O horizons was shown to be non-zero [64]. Interactions of a labile C substrate with the soil cannot be easily predicted as they involve a number of complex chemical mechanisms that operate at various timescales [30]. As the binding sites in soil are limited, sorption saturates at high solution concentration. The partition between sorption to the solid and liquid phase was predicted by Langmuir adsorption isotherms [40]. Isotherms were C compound specific since their molecular properties such as molar weight, polarity, and net charge dictate their sorption to the solid phase [65], with weakly charged anions of organic acids such as acetate binding considerably less than citrate [66,67]. Neutral C compounds such as sugars have a neutral charge and thus low sorption. Sorption was tracked as a separate pool per soil layer i for each C compound j where the total C bound to the solid phase $S_{abs\left(i,j\right)}$ (µmol cm^−3^) is the difference between the current pool size and the pool size estimated by the Langmuir adsorption isotherm for the solution concentration $C_{sol_{i,j}}$ (µmol cm^−3^):(11)$$dS_{abs\left(i,j\right)}=S_{j}\left(C_{soil\left(i,j\right)}\right)*\rho_{i}*vol\left(i\right)-S_{abs\left(i.j\right)},$$
where $S_{j}$ is the predicted amount of C compound j absorbed to the soil per g of soil (µmol g^−1^), $\rho_{i}$ is the soil bulk density g cm^−3^, and vol(i) is the volume of soil cylinder i. Sorption for C compound j is estimated [68] based on the maximum sorption $S_{max}$ and the affinity coefficient $K_{s}$ for each compound by:(12)$$S_{j}\left(C_{soil\left(i,j\right)}\right)=\frac{S_{max}K_{s}C_{soil\left(i,j\right)}}{\left(1+C_{soil\left(i,j\right)}K_{s}\right)}.$$

### 4.4. Mineralization Modeling Approach (M)

Mineralization of low molecular weight (LMW) C compounds proceeds at rates orders of magnitude greater than litter pools [69]. Half-lives for amino acids in aerobic soil are consistently reported as less than six hours [70]. Although anaerobic mineralization of litter is intrinsically slower than with the presence of oxygen, evidence from lake sediments [71], intertidal sediment [44], and marine glacial basin [72] indicate that mineralization rates of simple compounds like glucose vary greater between sites than between anoxic and oxic conditions. For example, glucose mineralization rates for anaerobic rice paddy soil were reported to have half-lives between 0.04–3.15 h [73], while half-lives of 1.10–46.2 h for salt marshes were reported [74]. These rates are comparable to those reported for agricultural grasslands [41], podsoils [75], and various topsoil types [76]. Differences between aerobic and anaerobic mineralization rates may be inconsequential since *E. vaginatum* roots contain aerenchyma tissues that ventilate the adjacent soil, likely sustaining oxic conditions up to one cm from the root surface [77]. Hence, this study assumes no difference between aerobic and anaerobic mineralization half-lives. The approach employed here simulates mineralization of available solutes in each layer as a first order process (similar to [32]), with base rates modified according to:(13)$$\frac{dC_{sol\left(i,j\right)}}{dt}=-k_{j}C_{sol\left(i,j\right)}f_{mb}\left(d\right)f_{T}\left(d\right),$$
where $k_{j}$ is the base mineralization rate of C compound j, $C_{sol\left(i,j\right)}$ is the pool size of C compound j in soil layer i, and $f_{mb}\left(d\right)$ is a depth affect multiplier for peat microbial biomass, and $f_{T}\left(d\right)$ is a temperature effect mineralization modifier for layer i.

Mineralization rates are mediated by a consortia of microorganisms shaped by historical C deposition [78], activity [79], and biomass [80]. Upper soil layers have a higher root density and subsequent influx of labile C that supports a larger microbial biomass pool, and select for flora specializing in consuming exudate feedstocks [3]. Inverse relationships between depth and mineralization rates are common [81]. Glucose mineralization rates in salt marshes declined by 59% between the 0–5 cm and 20–25 cm depth intervals [74], whereas acetate mineralization declined by 100% between the 0–4 cm and 18–22 cm depth intervals [82]. Microbial biomass strongly influences peatland mineralization rates [26]. In Mer Bleue, microbial biomass declines to ~53% of the surface peak value at 25 cm depth, concomitant with a decrease of 26–38% in litter decomposition rates [83], and organic acid half-lives were 2–6 times longer at 30–45 cm than 0–15 cm soil depth intervals [40]. Mineralization rates were assumed to decline with microbial biomass which declines exponentially with depth. The microbial biomass modifier $\;f_{mb}$ is given by the following equation:(14)$$f_{mb}\left(d\right)=e^{\left(i_{mb}+j_{mb}*d\right)},$$
where $i_{mb}=0.0287,$ and $j_{mb}=-0.024$ were fit with the nls package in R using the normalized microbial biomass [26].

The temperature modifier was the commonly employed Q10 function [43,84,85,86], utilizing values from published research on Mer Bleue. Q10 values for recalcitrant C typically range from 2.7 [87] to 6 [88], whereas labile peat displays less temperature sensitivity. Little is known about the effects of temperature on LMW C mineralization. Labile litter and peat was modeled with a smaller Q10 value of 2.3 [44,89] and estimated a Q10 of 2.0 for three amino acids over a 1–30 °C range. Mineralization temperature sensitivity was modeled by:(15)$$f_{T}\left(d\right)=Q10^{\left(T_{i}\left(d\right)/10\right)}\;T_{i}>0\;{}^{\circ}C,$$
where Q10 = 2.3, $T_{i}$ is the temperature at soil layer i. Soil temperature was modeled as a logarithmic function with depth based on the growing season daily average soil temperature (day of year 150–250) recorded for a Hollow at Mer Bleue by soil temperature probes at 10, 20, 40, 60, 80, 150, 250 cm depths for the years 1998–2010. The hollow site was chosen as features less interannual temperature variability. The soil temperature with depth relationship is an approximation of the summer average temperatures and does not account for soil temperature dynamics, which produce larger swings in temperature at shallower soil depth [43]. The soil temperature was simulated as per [43] by:(16)$$T_{i}\left(d\right)=\;x_{T}-y_{T}ln\left(d\right),$$
where $x_{T}=17.8$ and $y_{T}=2.12$.

### 4.5. Numerical Analysis

Numerical analysis was conducted in two parts. Accumulation gradients were first estimated based on a C compound specific parameterization. Second, a global sensitivity analysis was conducted for non-polarized exudate compounds (sugars and amino acids). As insufficient sample size or improper sample distribution can hinder the reliability of a sensitivity analysis [90], the parameter space was based on the range of parameters found in the literature for said class (Table 1). All parameters were varied simultaneously [91] utilizing a Sobel sequence [92] to generate a uniform distribution in the multivariate parameter space [93]. The contribution of any parameter to the variance in model outputs was evaluated in R 2.9.0 and the sensitivity package for R [94].

For both analysis, accumulation gradients were simulated using 500 soil annuluses representing a potential rhizosphere radius of 10 cm (0.2 mm width per annulus). The potential rhizosphere radius was set high to ensure the model reached equilibrium before the last annulus. Exchanges between soil annuluses were set simulated with a one second time step. The accumulation gradient was assumed to have reached equilibrium when the relative change in solute concentration between time steps was less than 0.00001%. For each soil annulus, model outputs include the soil solute concentration, the solute concentration bound to the solid phase, the portion of solute mineralized per time step, the net flux between adjacent soil annuluses per time step. The rhizoplane concentration, the time to reach equilibrium, the rhizosphere extent are derived from the model outputs.

## 5. Conclusions

This simulation model has produced estimates of exudation rates that are in line with previous reports, while demonstrating that exudation rates for C compounds regulated by influx do vary with depth on account of greater recapture of accumulated exudates. The insights provided by this model help provide a better understanding of exudate regulation in peatlands. The quantity and composition of exudates may help determine the labile C pool available to the methanogen community, while understanding the size of the rhizosphere has downstream implications for whether exudates escape the oxic zone around the root. These two factors are critical for assessing the flux magnitude of exudates to the microbial community.

The model results and sensitivity analysis reveal that depth was one of the top controls on root exudation. Depth also controlled the relative influence of the controls, with uptake rate and the Michaelis–Menten coefficient increasing in deeper soil. The simulated effects of soil properties on exudation were found to be considerable, resulting in rates that differed by a factor of two. Based on these results, upscaling schemes attempting to translate single root measurements to the whole root system should account for soil property induced gradients in net exudation. Furthermore, peatland models may need to consider vertical variability in exudation rates when simulating soil priming effects, methanogenesis production and transport, and C sequestration. Overall, these findings suggest that deeper roots are more competitive against the microbial community for re-uptake of exudates. However, root density has also been suggested to increase root exudation re-uptake, and influx dynamics have yet to be investigated. Further research is needed to validate soil property influences on exudation rates and to understand the functional significance of this finding in the many and important processes in the belowground processing of C.

## Figures and Tables

**Figure 1 plants-10-00106-f001:**
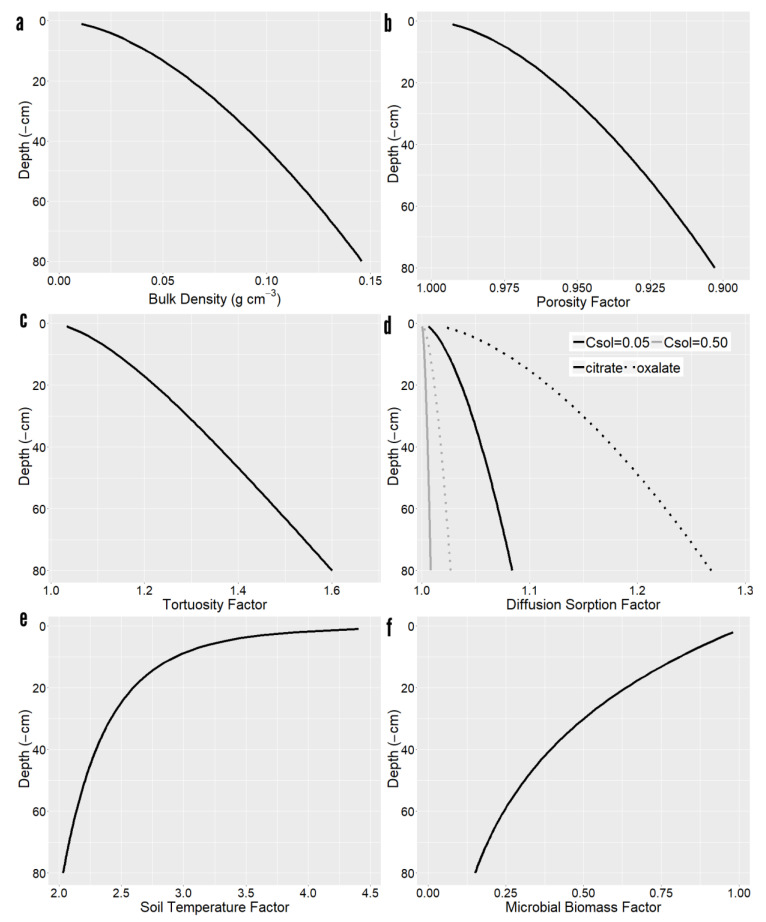
Depth dependence of model parameters. (**a**) Bulk density. (**b**) Porosity factor. (**c**) Tortuosity factor. (**d**) Diffusion sorption for citrate and oxalate at 0.05 and 0.5 µmol cm^−3^ concentration. (**e**) Soil temperature factor. (**f**) Microbial biomass factor.

**Figure 2 plants-10-00106-f002:**
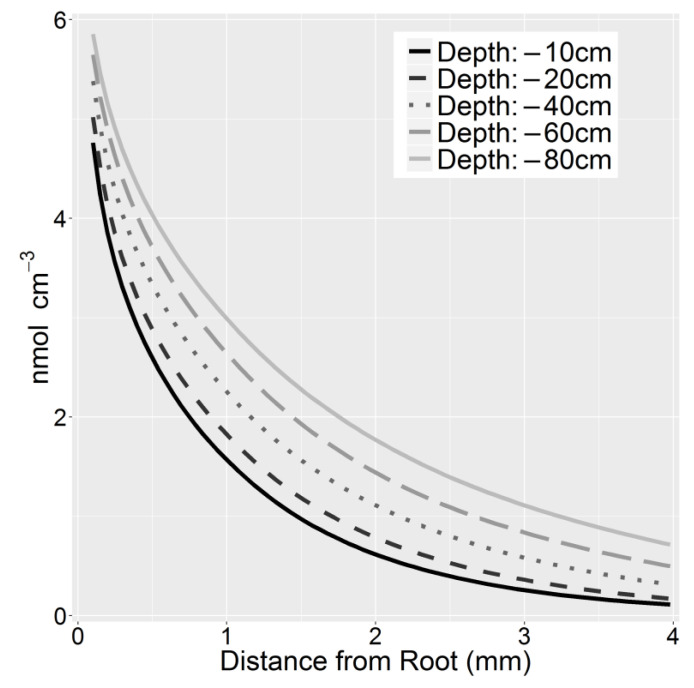
Rhizosphere accumulation gradients for glucose as a function of soil depth.

**Figure 3 plants-10-00106-f003:**
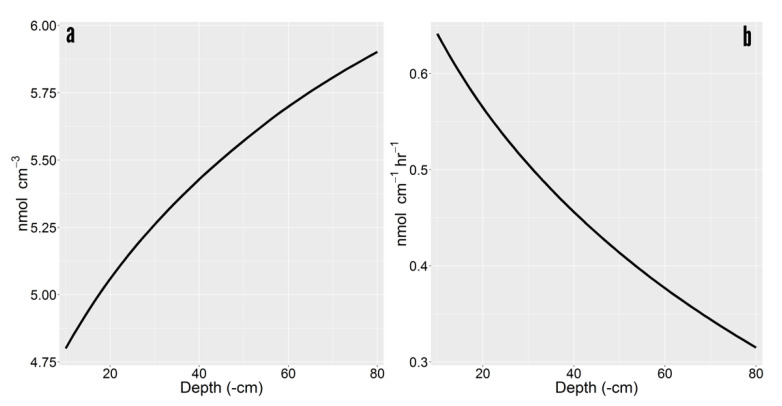
Depth induced exudation variability of glucose. (**a**) Concentration of accumulated exudates on the rhizoplane. (**b**) Net exudation rate as a function of depth.

**Figure 4 plants-10-00106-f004:**
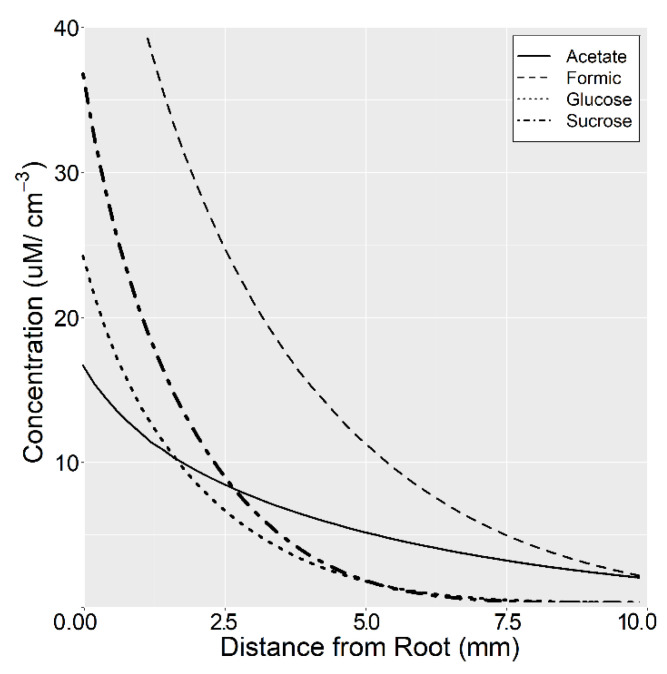
Simulated rhizosphere accumulation gradients for select low molecular weight (LMW) C compounds at −40 cm depth.

**Figure 5 plants-10-00106-f005:**
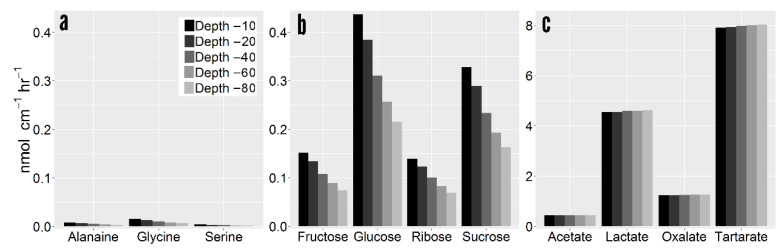
Net root exudation per compound grouped by class. (**a**) Amino acids. (**b**) Sugars. (**c**) Organic acids. Note the scale of sugars varies from amino acids and sugars.

**Figure 6 plants-10-00106-f006:**
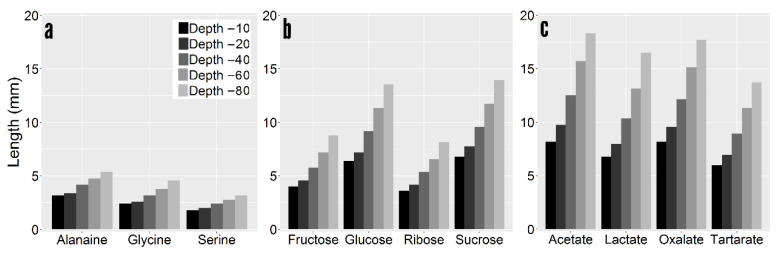
Size of the rhizosphere at 0.01 nmol cm-3 threshold per LMW C compound. (**a**) Amino acids. (**b**) Sugars. (**c**) Organic acids.

**Figure 7 plants-10-00106-f007:**
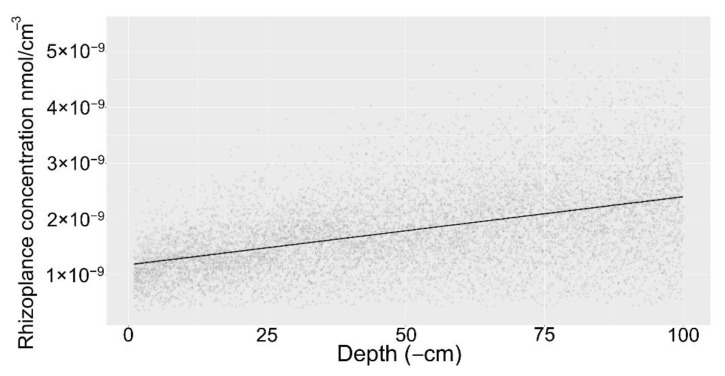
Global sensitivity analysis of root exudate model for sugar and amino acid exudation indicating the rhizoplane concentration as a function of depth. Grey indicates the simulation results, black line represents a linear regression model.

**Figure 8 plants-10-00106-f008:**
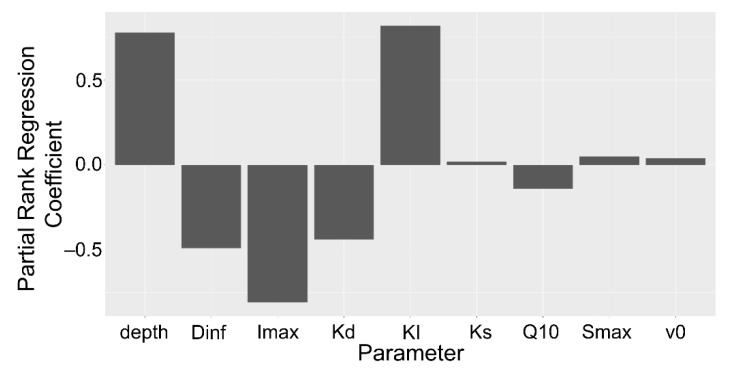
Barplot of partial rank correlation coefficients of model parameters for rhizoplane concentration. Parameters: Diffusion coefficient of solute in H2O (Dinf), maximum uptake rate (Imax), mineralization rate (Kd), Michaelis–Menten coefficient (KI), sorption affinity coefficient (KS), Q10 temperature coefficient (Q10), maximum sorption (Smax), volume of water entering the root (v0).

**Figure 9 plants-10-00106-f009:**
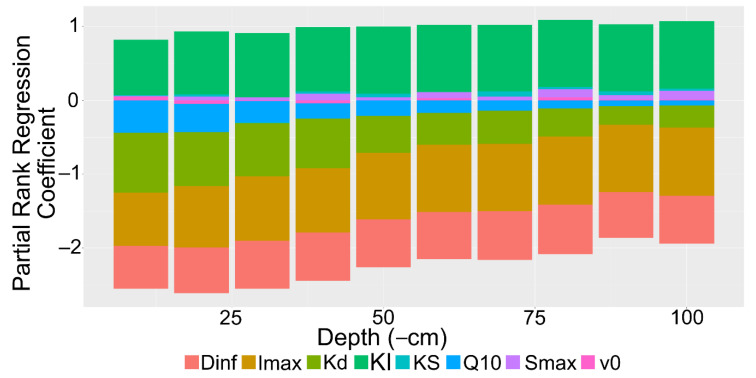
Stacked barplot of partial rank correlation coefficients of model parameters for rhizoplane concentration by depth interval. All simulations within 10 cm depth intervals were grouped. Parameters: Diffusion coefficient of solute in H2O (Dinf), maximum uptake rate (Imax), mineralization rate (Kd), Michaelis–Menten coefficient (KI), sorption affinity coefficient (KS), Q10 temperature coefficient (Q10), maximum sorption (Smax), volume of water entering the root (v0).

**Figure 10 plants-10-00106-f010:**
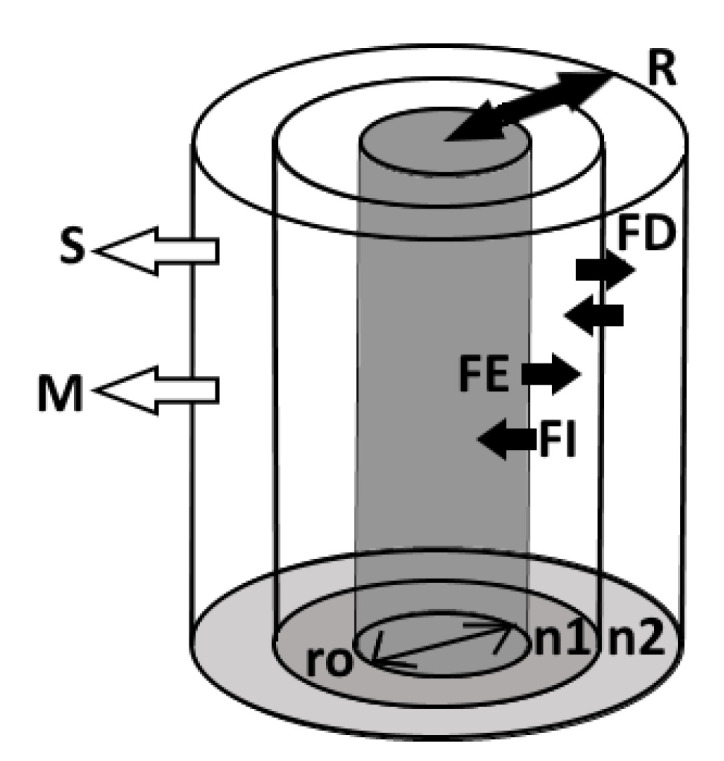
Model structure. The inner grey cylinder represents the root. The root is surrounded by n soil cylinders with the first cylinder, n1 adjacent to the root. The closed arrows of $F_{D}$, $F_{E}$, $F_{I}$ represent the fluxes from diffusion, exudation efflux, and exudation influx. The open arrows S, M represent the soil solution concentration lost to sorption and mineralization. $r_{0}$ is the root radius, while R is the width of the rhizosphere.

**Table 1 plants-10-00106-t001:** Model parameters and range utilized in this study.

Symbol	Definition	Unit	Range	Ref.
Solute Parameters (* range is solute specific)				
*Z*	Charge of solute in solution *	−integer	−1 to −3	
$D_{\infty }$	Diffusion coefficient of solute in H_2_O *	cm^2^ s^−1^	0.52–1.28	[95,96,97]
$S_{max}$	Maximum sorption *	µmol g^−1^	0.12–19.98	[30,51,68,75,89,98]
$K_{s}$	Sorption affinity coefficient *	Unitless	0.145–4.3	[30,51,68,75,89,98]
*kd*	Mineralization rate *	hr^−1^	0.15–2.35	[41,64,68,69,95,96]
Root Parameters (* Solute specific)				
l	Root length	cm	Constant	
r0	Root radius	cm	Constant	
Ccyto	Root cell cytoplasm concentration *	µmol cm^−3^	0.5–40	[99]
Perm	Membrane permeability coefficient *	cm hr^−1^ × 10^−4^	1.15–4.32	[54]
$I_{max}$	Maximum uptake rate *	µmol cm^−1^ hr^−1^	0.006–0.07	[24,54]
$K_{I}$	Michaelis-Menten coefficient *	µmol cm^−3^	0.002–5.9	[54,100]
$E_{m}$	Membrane potential	mV	Constant	[83]
v0	Volume of water entering the root	cm^3^/s	5.0 × 10^−10^–5.0 × 10^−6^	[101]
Soil Parameters (all vary by depth)				
τ(d)	Tortuosity factor	unitless	2–3	[61]
ρ(d)	Bulk density	g cm^−3^	0.0107–0.146	[58]
$f_{mb}\left(d\right)$	Microbial biomass factor	unitless	0–1	[102]
$f_{T}\left(T\right)$	Soil temperature factor	unitless	1.9–4.4	
T(d)	Soil temperature	Celsius	8–17.8	
*ϵ*(*d*)	Soil porosity	cm^3^/cm^3^	0–1	
R(d)	Sorption retardation factor per solute	unitless	>1	

## Data Availability

The data presented in this study are available on request from the corresponding author. The data are not publicly available due to the costs of online hosting of files of this magnitude.

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
