# Peer review of "Modeling Root Exudate Accumulation Gradients to Estimate Net Exudation Rates by Peatland Soil Depth"

_plants, 2021, doi:10.3390/plants10010106_

Round 1

Reviewer 1 Report

This manuscript is focused on the modelisation of root exudates accumulattion gradients according to the depth. This is new.

The objectives are clear and well introduced. The models and the mathemical equations used in the manuscript are well described and explained.

I have no special remarks about this manuscript except :

  • it's a little bit long. I think that you could reduce some parts of the manuscript. I needed a high concentration when reading it.
  • on the contrary, the conclusion is too short. Please add some informations especially about the use of the results for researchears working on root exudates. Your conclusion is too theoritical.

Reviewer 2 Report

For once, I don't have any major suggestions for the reviewers. The paper is well-written, data is well presenting and findings well interpreted. I think this paper will generate significant citations in future years.

Reviewer 3 Report

The manuscript by Procter and He demonstrated the regulation of root exudation via soil properties considering the root depth. This model revealed differences of rhizosphere size for various primary metabolites depending on the depth. Overall, the manuscript is well written, and this modelling approach could contribute to our understanding of the rhizosphere shaped by the exudation of metabolites from roots. The authors focus on primary metabolites such as amino acids, sugars, organic acids, but root also secretes secondary metabolites which function in shaping rhizosphere microbiota and in mediating biological communication. Root also secrete proteins to rhizosphere. I would recommend the authors to cite the review article of YakovKuzyakov (https://doi.org/10.1016/j.soilbio.2019.05.011) and discuss the contribution of protein and other metabolites. Rhizosphere modelling for secondary metabolite is also recently published and also to be cited (https://doi.org/10.1111/pce.13708).

Reviewer 4 Report

Summary:

The authors present a single root model, in which they couple concentration specific exudation with depth dependent soil properties. They apply this model on a peatland ecosystem and investigate the impact and significance of gradients in soil properties on the concentration of diverse exudates in the rhizosphere of E. vaginatum.

Broad comments:

  • The overall quality of the study is very sound in my opinion. Structure and methodology are clear and results are novel to my knowledge. I am not sure why a peatland ecosystem was chosen instead of an agricultural soil for which the results would be more meaningful and interesting for a broader audience.
  • There are several (volatility) errors in the paper that I think must be addressed: Often it is not clear where equations come from; a reference should always be given. It is often said that something declines x% - it is not clear to me if it drops by x% or to x% - please specify. When depths are given, sometimes a ‘–‘ sign is used and sometimes not, please be consistent. When soil depth is specified, it is mostly done like in line 446: ‘[…] declined between 0-5cm’. I think, it would be much clearer to write it out as ‘declined between a soil depth of 0 and 5 cm.’ I would not use e.g. ‘2-6x longer’, but rather write that out as ‘2-6 times longer’. Figures are not always numbered in the order of appearance. Sometimes, parameter symbols are use in the text, sometimes, the parameter names are written out – please be consistent.
  • In some parts, there are some language issues (was/were,…). – I am not a native speaker though and I guess you are, so maybe these are also just volatility errors? Please revise them.

Specific comments:

  • As a reader, I would prefer to have the Material and Methods section before the Results section, but that is up to you.
  • It’s Michaelis-Menten, not Michaelis-Menton
  • Figure 3: For which substance did you create this plot? (I am assuming glucose), but why did you choose that one?
  • Figure 8: I would appreciate if you could specify in the caption what the symbols mean
  • References are missing in equations [10], [12], [13], [16]
  • I could not find equation [4] and [14] in the given references, please provide the correct references
  • Line 101-102: revise sentence
  • Line 128: it’s rather logic that non-sorbing compounds do not have any influence on sorption??
  • Line 136: fig 2 / fig 3??
  • Line 159: how many?
  • Line 164: higher than where?
  • Line 230-231: Please use either sysmbols or parameter names, but be consistent
  • Line 247-248: please revise sentence
  • Line 252: weight to root ratio and molar mass: please insert reference
  • Line 258: why 50 m per square meter? Where is this value from?
  • Line 260: NPP not in table 1
  • Line 271: which variables, please specify
  • Line 272: not clear why you write about observation windows
  • Line 355: why is the reference written out?
  • Line 409-412: please rephrase
  • Line 474: what is the ‘best available information’?
